# Exploring Anticitrullinated Antibodies (ACPAs) and Serum-Derived Exosomes Cargoes

**DOI:** 10.3390/antib14010010

**Published:** 2025-01-26

**Authors:** Mohammed A. Alghamdi, Sami M. Bahlas, Sultan Abdulmughni Alamry, Ehab H. Mattar, Elrashdy M. Redwan

**Affiliations:** 1Department of Biological Sciences, Faculty of Science, King Abdulaziz University, P.O. Box 80203, Jeddah 21589, Saudi Arabia; 2Laboratory Department, University Medical Services Center, King Abdulaziz University, P.O. Box 80200, Jeddah 21589, Saudi Arabia; 3Department of Internal Medicine, Faculty of Medicine, King Abdulaziz University, P.O. Box 80215, Jeddah 21589, Saudi Arabia; 4Immunology Diagnostic Laboratory Department, King Abdulaziz University Hospital, P.O. Box 80215, Jeddah 21589, Saudi Arabia; 5Centre of Excellence in Bionanoscience Research, King Abdulaziz University, Jeddah 21589, Saudi Arabia; 6Therapeutic and Protective Proteins Laboratory, Protein Research Department, Genetic Engineering and Biotechnology Research Institute, City for Scientific Research and Technology Applications, New Borg EL-Arab, Alexandria 21934, Egypt

**Keywords:** rheumatoid arthritis (RA), anticitrullinated protein antibodies (ACPAs), risk factors, citrullinated fibrinogen (cFBG), exosomes, serum-derived exosomes

## Abstract

Background: Autoantibodies such as rheumatoid factor (RF) and anticitrullinated protein autoantibodies (ACPAs) are useful tools for rheumatoid arthritis (RA). The presence of ACPAs against citrullinated proteins (CPs), especially citrullinated fibrinogen (cFBG), seems to be a useful serological marker for diagnosing RA. RA patients’ sera were found to be enriched in exosomes that can transmit many proteins. Exosomes have been found to express citrullinated protein such as cFBG. Objective: We conducted this study in two stages. In the first phase, we aimed to evaluate the association between autoantibodies and risk factors. In the next step, ACPA-positive serum samples from the first phase were subjected to exosomal studies to explore the presence of cFBG, which is a frequent target for ACPAs. Methods: We investigated the autoantibodies in one hundred and sixteen Saudi RA patients and correlated with host-related risk factors. Exosomes were extracted from patients’ sera and examined for the presence of cFBG using monoclonal antibodies. Results: The study reported a high female-to-male ratio of 8:1, and seropositive RA (SPRA) was more frequent among included RA patients. The frequency and the levels of ACPAs were similar in both genders. Autoantibodies incidences have a direct correlations with patient age, while the average titers decreased as the age increased. Further, the highest incidence and levels of autoantibodies were reported in patients with RA duration between 5 and 10 years. Smoking and family history have no impact on autoantibody, except for ACPAs titers among smokers’ RA. Our analysis of serum exosomes revealed that about 50% of SPRA patients expressed cFBG. Conclusions: The female-to-male ratio is 8:1, which is higher than the global ratio. We can conclude that patients’ age and disease duration contribute to the autoantibodies, particularly RF and anti-MCV, whereas smoking and family history had no effects on autoantibodies. We detected cFBG in all exosomes from SPRA patients; thus, we suggest that the precise mechanism of exosomes in RA pathogenesis can be investigated to develop effective treatment strategies.

## 1. Introduction

Rheumatoid arthritis (RA) is a systemic autoimmune disease of unknown etiology. RA is the most common chronic disorder, with a global prevalence incidence of 0.5–1% [1]. According to several systematic review and meta-analysis, the global prevalence for RA ranges from 0.24 to 1%, and it is more common in females than males, with an estimated ratio about 3:1 [2,3,4]. In Saudi Arabia, there are no adequate data available about the exact prevalence of RA. According to a few past studies, the prevalence of the disease in Saudi Arabia is about 0.1−0.22%, with three to four times higher rates in women than men [5,6,7,8]. Citrullination is a post-translation modification (PMT) process related to human physiology and some pathological diseases. In inflammatory diseases such as RA, MS, and PsA, citrullinated peptides (CPs) have been found to trigger antibodies against such modified proteins [9]. Autoantibody measures have been a constant companion for physicians managing RA patients, and their significance has grown over the past few decades. Autoantibody investigations include the measurement of rheumatoid factor (RF) and anticitrullinated peptide antibodies (ACPAs). ACPAs and RF improve diagnostic accuracy and are included in the 2010 ACR/EULAR criteria [10,11]. RF is commonly used as a diagnostic marker of RA, whereas anti-ACPAs, including anticyclic citrullinated peptide (anti-CCP) and antimutated citrullinated vimentin (anti-MCV), are being used as specific prognostic and diagnostic biomarkers for RA. Both anti-CCP and anti-MCV may exist in patients’ sera years before the appearance of clinical symptoms; thus, they can predict the early progression of RA [11,12]. However, RA could be classified as seropositive RA (SPRA) or seronegative RA (SNRA). SPRA refers to the presence of IgM-RF and/or ACPAs, whereas SNRA refers to the absence of these autoantibodies in confirmed RA. Based on clinical and laboratory evidence, seropositivity occurs in 60–80% of patients with confirmed RA [13].

Although the exact cause of RA is still unknown, the bulk of the evidence indicates that many factors might increase the risk of RA, including age, gender, and genetic, environmental, and metabolic factors. Smoking and infection represent the most environmental components that have potential roles in RA and can trigger the inflammatory process, especially in genetically predisposed individuals [14]. Studies using clinical and animal models have indicated that the etiopathogenesis of RA is influenced by infections caused by a variety of bacteria, including Epstein–Barr virus (EBV), *Proteus mirabilis* (*P. mirabilis*), *Porphyromonas gingivalis* (*P. gingivalis*), and mycoplasma [15].

Periodontitis is a bacterial infection of mucosal gum [16]. The biofilm of subgingival tissues in periodontitis lesions showed the presence of a variety of bacteria species, with the major bacteria causing the most aggressive destruction being *P. gingivalis* and *Aggregatibacter actinomycetemcomitans* (*A. actinomycetemcomitans*) and *Prevotella intermedia* (*P. intermedia*) [17]. A recent meta-analysis demonstrated that *P. gingivalis* is the most common periodontopathogenic bacterium that significantly correlates with RA [18]. This pathogen expresses bacterial peptidyl arginine deiminase (PPAD) enzymes that are calcium-independent, unlike human PADs, which require calcium for activation [19].

The combination of genetic and environmental factors is strongly associated with RA. The early diagnosis of RA is important in the treatment and prevention of worse stages [20]. Many epidemiologic studies indicate sex-related factors in RA risk, as two-thirds of RA patients are female. Therefore, it has long been thought that there are female-specific characteristics that increase the risk of RA [21]. A recent systematic review and meta-analysis investigated the association between gender type and serostatus, i.e., seropositive and seronegative. This meta-analysis found that men with RA are more likely to have seropositive RA than women. The results indicated that RF positivity in males was 16% higher than in females. Similarly, the analysis of ACPA seropositivity showed that the number of males who were ACPA-positive was 12% higher than females [22]. Many studies have elucidated that hormonal replacement therapy (HRT) and the use of oral contraceptives (OCs) for ≥7 years were effective against ACPA and RF development [23,24].

The frequency of RF and ACPA positivity is variable among different age groups. Some reports have revealed that seropositivity is more frequent in patients aged above 40 and up to 60 years [25,26], while it is less prevalent in other age groups, i.e., <40 and >60 years [26,27]. Considering disease duration, it has been found that there is no significant change in autoantibody existence in patients with early RA (<1 year) and established RA (>2 years). A slight rise has been reported in RF, anti-CCP, and anti-MCV [11]. However, another study found that anti-CCP and IgM-RF increased significantly after 5 years of duration [28]. This was supported by further studies that have found that anti-CCP frequency and level are exacerbated in the early onset of RA (≤1 year) and then decline gradually within 3–5 years. Afterward, anti-CCP increased significantly after a 5-year duration [29,30].

Extensive epidemiologic studies have demonstrated that the RA risk for smokers is two times higher than that of non-smokers, notably in male smoker RA patients [31,32]. Even in the absence of RA, smoking has long been linked to a positive RF [33,34]. Many studies have linked the presence of ACPA in smokers to the fact that smoking induces the citrullination process during lung inflammation. It has been found that there is a considerable correlation between smoking and anti-CCP concentration, whereas RF levels were comparable between smoker and non-smoker RA patients [35,36,37,38].

RA is highly heritable and, unfortunately, tends to run in families. According to various studies, 50–60 percent of RA cases are thought to be heritable [39,40]. Assessment of family history in autoimmune diseases may be considered before the identification of genetic factors [41,42]. The human leukocyte antigen (HLA) has essential roles in antigen presentation and immune response in RA. HLA-DRB1 carrying shared epitopes (SE) is a class II-HLA, and it is well established to be the strongest genetic risk factor for developing RA [43]. Recent studies have clarified the high expression of HLA-DRB1 SE on the immunocytes of RA patients. The HLA-DRB SE can bind and present the citrullinated protein and trigger the autoimmunity response in RA [44,45]. Considering the serostatus, early familiar and genetic studies found that family aggregation was higher in seropositive RA cases than in seronegative cases [46,47].

Interestingly, it has been verified that RA patients had higher levels of extracellular vesicles (EVs) than healthy individuals [48]. Furthermore, people with RA have been reported to have distinct exosome cargo, which may help with diagnosis [49]. Exosomes are the smallest and most well-studied class of EVs, with an average diameter of 30–150 nm and density of 1.13–1.19 g/mL; they have a spherical cup-like shape in EM images [50,51]. Exosome cargo includes proteins, lipids, nucleic acids, small molecules, and receptors. These nanoparticles have many critical roles in biological systems’ physiology, pathology, and therapy [52].

Exosomes encompass many proteins in their cargo, with the proteome of a typical exosome containing approximately 4400 proteins [53]. The most common proteins in exosomes are ESCRT proteins (Alix and tumor susceptibility gene 101 (TSG101)), heat-shock proteins (HSP70 and HSP90), and tetraspanin (CD9, CD63, and CD81). Distinctly, these exosome-enriched proteins are highly utilized as specific markers for exosomes [54]. The high stability of exosomes in the extracellular space enables them to carry their cargo far away to interact with distant cells [55].

The pathophysiology of RA has long been known to be significantly affected by infections and malignancy. A variety of immunological processes may be triggered by certain infections and cancers, which can affect the immune system through multiple pathways. There is mounting evidence that exosomes produced from saliva could be used as biomarkers for periodontitis. Toxins and other virulence factors can be transported by *P. gingivalis*’s outer membrane vesicles (OMVs), which also trigger the release of pro- and anti-inflammatory cytokines that stimulate osteoclasts, T and B lymphocytes, and neutrophils [56]. Furthermore, colon carcinogenesis, which occurs in people with inflammatory bowel infections, has the strongest correlation between malignancy and chronic inflammation, such as RA [57]. Exosomes can carry infectious agents and transport various genetic products, such as DNA and miRNA. Colon cancer and RA share genetic factors, such as the mutant form of the p53 gene that contributes to the inflammatory RA and tumorigenesis. Recent reports revealed that exosomal miR-1246 can transfer mutant p53 that alters macrophages into tumor-supporting macrophages [58,59].

The proteomic and immunological studies of immune cell-derived exosomes obtained from different body fluids revealed the significant role of these nanovesicles in the regulation and modulation of immune responses, including both immune suppression and immune stimulation [60]. Serum exosomes can transfer several proteins involved in RA pathogenesis and might be predictors of therapeutic responses. Serum exosomes exhibit antigen-presenting functions, as they can express HLA, especially HLA-DRB1 SE, which has a great affinity to citrullinated proteins [61]. A recent study has shown that B cells produce a high level of HLA-bearing exosomes [62]. Furthermore, proteomics of RA serum exosomes showed that exosomes could present the citrullinated peptides to effector CD8+ cells to release TNF-α and IFN-γ [63].

Citrullinated fibrinogen (cFBG) has been identified as a major autoantigen for ACPAs that play a role in RA pathogenesis. Increased levels of cFBG have been detected in the serum and synovium of RA patients and have been linked to joint damage and RA severity [64,65,66]. Analysis of exosomes derived from inflamed synovium led to the detection of cFBG on the exosome surface, which may trigger the production of ACPAs [67]. In our current work, we aim to explore the presence of cFBG in serum-derived exosomes. However, as far as we know, no studies have investigated cFBG in exosomes extracted from the serum of RA patients.

## 2. Materials and Methods

### 2.1. Study Population and Sample Collection

A set of 116 Saudi RA patients aged 18–70 years and 35 healthy controls (HC) aged 23–66 years were included in this study. The control group included demographically matched, healthy individuals chosen from the general population and the King Abdulaziz University Hospital (KAUH) staff. Patients were enrolled and investigated during their routine visits to the rheumatology clinic at KAUH. We selected the patients who met the inclusion criteria and agreed to participate in the study. The inclusion criteria required that patients be adults (≥18 years old) and have a confirmed RA according to the rheumatologist’s diagnosis or based on the fulfillment of the ACR 1987 criteria for RA.

The study was approved by the research ethics committee of the Faculty of Medicine at King Abdulaziz University (468-19), and all patients provided written informed consent for participation. Following consent, clinical and demographic data were obtained from the patients or their accompanying relatives. The parameters collected for the patients include age, gender, disease onset, disease duration, family history, smoking status, and occupation.

Serum samples were collected from 116 RA patients attending the rheumatology clinic at KAUH for routine appointments. One hundred and sixteen patients who fulfilled the inclusion criteria and consented to participate in the current study were subjected to blood collection. Whole blood was collected from the veins in the arms by drawing peripheral blood into serum separator tubes for laboratory testing. All collected samples were processed within 1 h of the collection and centrifuged for serum separation. A fraction of the serum was used for serological testing, while the other was subjected to exosome extraction and further exosomal analysis. The same procedure was applied to the healthy control group.

#### Demographic Characteristics of Patients and Healthy Controls

To improve the outcome of the study, we tried to find a matched control group. After patient group enrollment, we recruited thirty-five healthy control (HC) individuals who match the demographic characteristics of the patient group. Indeed, we chose individuals who are in good health and have no health issues. Further, we tested all HC for autoantibodies (RF, anti-CCP, and anti-MCV) and inflammatory markers such as CRP and ESR to ensure there was no abnormality that might be relevant to autoimmune conditions. Any participant who showed abnormal results was excluded from the study. Moreover, we ensured that the demographics of HC group were matched with those of the patients, such as gender, occupation, age, and age groups. In this study, we considered demographic variables that are known to be the most influential factors in RA. RA patients did not differ from healthy controls on demographic variables such as age, gender, and occupation. However, about 89% of patients were females compared to 80% females in the HC group, with a female/male ratio of 8:1 for the patient group.

The average age for all patients was 51.3 ± 11.5 years compared to 47 ± 11.6 years for the HC group. Further, symmetry in age distribution between both groups was considered for each age group. The vast bulk of participants were housewives: 69.8% of RA patients compared to 57.1% of HC. However, there were no significant differences (*p* > 0.05) between the patient and HC groups in any of the demographic characteristics assessed. Baseline demographic characteristics for patients and controls are summarized in Table 1.

### 2.2. Laboratory Analysis

Measurements of RF and ACPA (anti-CCP and anti-MCV) were performed in the accredited laboratory of immunology at KAUH. Exosome preparation, characterization, and the identification of citrullinated protein (cFBG) were performed in a research lab in the college of science at KAU.

#### 2.2.1. RF Testing

Serum was tested for RF-IgM by nephelometry assay on BN II system using an N Latex RF Kit (Siemens Healthcare Diagnostics, Marburg, Germany). All patients, HC, and QC samples were run according to the manufacturer’s instructions. The assay can detect RF in a range from 10 to 1500 IU/mL. The N Latex RF contains an immunocomplex of human immunoglobulin and anti-human IgG from sheep that aggregates with RF-IgM. The intensity of aggregation is proportional to RF concentration. The measurements were evaluated against the reference curve and reported in IU/mL; results ≥ 20 IU/mL were considered positive for RF.

#### 2.2.2. Anti-CCP and Anti-MCV Testing

The presence of anti-CCP and anti-MCV in the serum of all participants (patients and HC) was tested on an Alegria system (version: V 49, Orgentic, Mainz, Germany) using the anti-CCP and anti-MCV reagents (ORGENTEC Diagnostika GmbH, Mainz, Germany). Both reagents are ELISA-based assays for the quantitation of immunoglobulin G (IgG) class autoantibodies against citrullinated peptides in human serum or plasma samples. The anti-MCV includes specific epitopes purified from the native mutated citrullinated protein. All measurements were reported in U/mL, and the cut-off was 20 IU/mL. Any result ≥ 20 IU/mL was reported as positive for anti-CCP or anti-MCV.

### 2.3. Exosomal Study

Exosomes were extracted from serum samples using the commercial reagent Total Exosome Isolation for serum (Invitrogen, Vilnius, Lithuania) and subjected to morphological study and protein identification. Transmission electron microscopy (TEM) and dynamic light scattering (DLS) were used for morphological studies. Sodium dodecyl-sulfate polyacrylamide gel electrophoresis (PAGE) and Western blotting were used for the identification of exosome markers proteins and citrullinated fibrinogen (cFBG) in serum-derived exosomes.

#### 2.3.1. Exosome Isolation

Exosomes were extracted from the serum of RA patients (n = 116) and healthy individuals (n = 35) using the Total Exosome Isolation (TEI) kit from serum. All exosome preparation steps were performed in cold conditions. Initially, serum samples were centrifuged at 2000× *g* for 30 min to remove cells and debris, and the supernatant was transferred to a new tube. Then, 30 µL of TEI reagent was added to 150 µL of serum and mixed by vortex until a homogenous solution with a cloudy appearance formed. The mixtures were then incubated at 4 °C for 30 min. After incubation, mixtures were centrifuged at 10,000× *g* for 10 min at 4 °C The supernatants were carefully discarded, and the pellet containing the exosomes was subjected to three cycles of washing with cold 1× PBS (Gibco, Waltham, MA, USA). Then, the pellet was resuspended entirely in 75 µL of cold 1× PBS. Each pellet suspension was divided into two equal volumes and kept at −80 °C for further analysis: one part for exosomes morphology and the other for exosomes protein identification.

#### 2.3.2. Exosome Size Measurement by Dynamic Light Scattering

Exosome size measurements were analyzed using the Zetasizer Nano ZS ZEN3600 instrument (Malvern Panalytical, Malvern, UK). Exosome preparations from the original suspensions (non-lysed) were diluted 1:100 with sterile/filtered 1× PBS to a total volume of 1 mL. However, further dilutions were occasionally required to obtain an ideal concentration and avoid aggregation. The suspensions were then mixed by vortex for 1 min, immediately poured into Kuvetten Cuvettes (SARSTEDT), and placed in the instrument for measurements. The temperature was set at a stable 25 °C, the analyzer was allowed to equilibrate the temperature for 120 s, and each sample was run for three consecutive measurements.

#### 2.3.3. Exosome Characterization by Transmission Electron Microscopy

In this study, we used the high-resolution transmission electron microscopy (HRTEM) Titan CT from FEI Company to characterize the morphology of serum-derived exosomes using the negative staining technique. Briefly, freshly isolated exosomes (≈3 µg) in PBS suspension were fixed at a ratio of 1:100 with 2.5% glutaraldehyde. After 10 min of fixation at room temperature, a 10 µL drop of exosomes was gently loaded onto Formvar/carbon-coated 200 mesh copper grids and allowed to adsorb for 5 min at room temperature. For negative staining, the absorbed exosomes were stained with a 10 µL drop of the 1% phosphotungstic acid (PTA) for 5 min at room temperature. The excess solution was wiped off by filter paper and then allowed to air dry. TEM analysis was performed at an accelerating voltage of 80 kV. Digital images were visualized using a Gatan 2 k × 2 k slow-scan charged coupled device camera, and 20 fields were captured for each sample.

#### 2.3.4. Quantitation of Exosomal Protein Content

Prior to electrophoretic analysis and immunoblotting, each exosome sample was lysed with 1× radioimmunoprecipitation assay (RIPA) buffer (Merck Millipore, Temecula, CA, USA) containing phenylmethylsulfonyl fluoride (PMSF) protease inhibitor (Sigma-Aldrich, St. Louis, MO, USA) at 1 mM final concentration. The protein content of exosome samples was measured by the Pierce Detergent Compatible Bradford Assay Kit (Thermo Scientific, Rockford, IL, USA). A set of albumin standards was prepared to determine the standard curve. Briefly, 10 µL from each exosome and standard sample was mixed with 300 µL of Pierce detergent-compatible Bradford reagent, followed by 30 min of incubation at RT. The absorbance of each standard and sample was measured spectrophotometrically in duplicate at 595 nm using a BioTek ELx800 plate reader (Agilent BioTek, Winooski, VT, USA). All measurements were plotted against the BSA standard curve to determine the protein concentration.

#### 2.3.5. Sodium Dodecyl Sulfate-Polyacrylamide Gel Electrophoresis (SDS-PAGE)

Using the Hoefer™ Mini Vertical Electrophoresis Unit (Hoefer, San Francisco, CA, USA), two hand-cast gels were prepared: 4% stacking gel and 12% resolving gel. All exosome samples were separated under denatured and reducing conditions. Cold exosome lysates were mixed with 4X sample loading buffer containing β-Mercaptoethanol. The mixtures were then denatured by heating at 95 °C for 5 min. About 10 µg of exosome lysate was loaded into each gel well, and the PageRuler Plus Prestained Protein Ladder (Thermo Fisher, Waltham, MA, USA) was loaded in a separate well as a protein molecular weight marker. The current was set at 70 volts for 50 min and then at 120 volts for 2 h. After the electrophoresis was complete, the gel was washed with distilled water and stained with Coomassie blue stain for 15 min on the rocking platform. Finally, the gel was washed and destained with a destaining solution.

#### 2.3.6. Western Blot Analysis

Using Mini-PROTEAN^®^ Tetra Handcast (Bio-Rad, Hercules, CA, USA), exosome lysates were separated by 12% SDS-PAGE. Exosome lysates were mixed with 4X non-reducing sample loading buffer and heated at 95 °C for 5 min. Approximately 35 µg of each lysate was loaded into each gel well simultaneously with a protein molecular weight marker (Thermo Fisher). The electrophoresis was carried out at 70 volts for 50 min and then at 120 volts for 2 h. Then, exosome samples were transferred onto PVDF membrane (Bio-Rad, Hercules, CA, USA) using Mini Trans-Blot Module Systems (Bio-Rad, Hercules, CA, USA) under constant current at 100 V for 1 h. The blotted membranes were washed in 1× Tris-buffered Saline, tween 20 (TBST-20) for four cycles and blocked in a 3% BSA (Sigma-Aldrich) for 1 h. After clocking, the membranes were probed with primary antibodies dissolved in blocking buffer: CD9 monoclonal antibody (1:1000, Invitrogen-Fisher Scientific, Waltham, MA, USA) and citrullinated fibrinogen monoclonal antibody (1:1000, Cayman Chemical, Ann Arbor, MI, USA). Membranes were incubated with primary antibodies for 1 h at room temperature, followed by four washing cycles with 1× TBST. Peroxidase-conjugated AffiniPure Mouse Anti-Human IgG (H + L) antibody (Jackson ImmunoResearch, West Grove, PA, USA) was used as a secondary antibody after being diluted 1:5000 in the blocking buffer. Membranes were incubated with secondary antibodies for 1 h at room temperature, followed by four washings with 1× TBST. Proteins were then visualized using chromogenic DAB substrate (Fisher Scientific, Waltham, MA, USA).

### 2.4. Statistical Analysis

The statistical program GraphPad Prism (Prism 10 for Windows, Boston, MA, USA) was used to perform data analysis and to calculate percentages, means, standard deviations, minimums, and maximums. Proportions were calculated for categorical variables, and the chi-square test was utilized to evaluate and compare the statistical significance in frequencies between two sets of categorical data. Means and standard deviations were calculated for quantitative data. One-way ANOVA followed by Tukey’s post hoc test was employed for comparing the means among multiple groups of data. An unpaired *t*-test was used to compare the differences in means between two groups. All significant tests were two-tailed, with a reliability of 95%. *p*-values less than 0.05 were considered statistically significant.

## 3. Results

### 3.1. Demographic and Laboratory Investigations of RA Patients

A total of 116 Saudi patients with confirmed RA were investigated in this study; most of them were female (89.7%). Patients were grouped into three categories according to age: <40 years, 40–50 years, and >50 years. The average age of the total population was 51.3 ± 11.5 years; female and male groups had similar mean ages of 51.3 ± 11.6 and 51.7 ± 11.3, respectively. However, there was no statistically significant difference (*p* > 0.05) between age groups for both genders. A large segment of the patients was above 50 years of age (55.2%); of the 104 female participants, 53.8% were above 50 years of age compared to 66.7% of the male group. The disease duration was evaluated, and the patients were grouped into three levels of duration (<5, 5–10, and >10 years). For all patients, there was no statistically significant difference (*p* > 0.05) between disease durations. The female patients had an even distribution of RA duration, while 58.3% of the male patients had a disease duration between 5 and 10 years. However, the average duration for all subjects was 8.4 ± 6 years, while the average age of RA onset for both genders was 43 years old.

In addition, risk factors associated with RA, such as smoking and family history of RA, were investigated. Patients were divided into those who never smoked and those who had smoked in the past or were current smokers. Patients were considered to have a family history if they had at least one first-degree relative (FDR) with RA. However, about 18.1% of the total population were smokers, and 21.6% had a family history of RA. In terms of laboratory testing, serum samples collected from the patients were investigated for autoantibodies (RF, anti-CCP, and anti-MCV).

Of the 116 RA patients, 37.1%, 54.8%, and 48.3% tested positive for RF, anti-CCP, and anti-MCV, respectively. Hence, we next aimed to assess the relationship between laboratory findings and demographic characteristics. All demographic and laboratory characteristics of the RA patients are presented in Table 2.

### 3.2. Autoantiboies Investigations

#### 3.2.1. Association Between Autoantibodies and Gender Types

We investigated the relationship between positive antibodies and gender types based on laboratory results in terms of prevalence and concentration (Table 3). The seropositivity for all patients was defined for each of the tested antibodies (RF, anti-CCP, and anti-MCV). We found that anti-CCP was the most frequent antibody for both genders, followed by anti-MCV and RF in order. The laboratory results showed that 43 (37%) patients from both genders tested positive for RF, 63 (54%) patients were positive for anti-CCP, and 56 (48%) had a positive anti-MCV antibody.

The frequency of each antibody was compared between males and females using the chi-square test. The presence of positive antibodies was higher in males than females for each type of antibody. About 53% of females were positive for anti-CCP compared to 67% of males. RF was positive in 36% of females and 50% of males, while positive anti-MCV was found in 47% of females and 58% of males. However, the occurrence of positive antibodies was 1.2–1.4-fold higher in the male group than in the female group. Overall, there was no statistically significant difference (*p* > 0.05) in the prevalence of each antibody in males and females (Figure 1A).

Figure 1B shows the serological level of each antibody for both gender types. The quantitative data, which are given as means ± standard deviations (SD), were compared using an unpaired *t*-test. The concentrations (mean ± SD) for anti-CCP were comparable between females (547.8 ± 437.8) and males (697.1 ± 420.6), and the anti-MCV titer was (415.2 ± 406.6) for females and (589.2 ± 383.8) for males, showing no significant differences (*p* > 0.05). However, the average level of positive RF in the male group (356.4 ± 512.8) was higher than that in the female group (196.8 ± 236.6), showing a significant statistical difference (*p* = 0.004).

#### 3.2.2. Relation Between Autoantibodies and Patients’ Age

The prevalence and levels of RF, anti-CCP, and anti-MCV were also correlated with the age of RA patients. In line with many studies, RA patients were divided according to their age into three groups: <40, 40–50, and >50 years. As estimated before in Table 2, the average age for all RA patients was 51.3 ± 11.5 years, with 55.2% over 50 years of age. However, patients with positive results for antibodies had an average age of 53.3 ± 10.8 years. We investigated the association of each antibody with patient age, and we found that the incidence of positive antibodies gradually increased with age. The frequencies of anti-CCP and anti-MCV were 1.3–1.5-fold higher in RA patients above 50 years of age than those under 40 years, while the prevalence of RF increased by ~2.4-fold between the same groups (Table 4).

The incidence of anti-CCP among the three age groups was similar (*p* > 0.05). However, the positivity of anti-MCV was comparable except between <40 years (35%) and >50 years (53%), showing a statistically significant difference (*p* = 0.02). Moreover, RF prevalence in patients above 50 years of age (47%) was higher than in those less than 40 years of age (20%) and those between 40 and 50 years of age (28%), showing a significant difference (*p* < 0.001 and *p* = 0.008, respectively) (Figure 2A).

We further correlated the concentration of each antibody with the age of the patients. Overall, the level of each antibody decreased as the age increased, except for anti-MCV in patients >50 years. The average concentration (U/mL) for all antibodies was calculated in each age group and compared by one-way ANOVA followed by Tukey’s post hoc test. The highest levels of RF, anti-CCP, and anti-MCV were reported in RA patients under 40 years of age. The titer of RF (mean ± SD) in the group of <40 years was 2.3–2.6-fold higher than other age groups. It showed significant differences when compared to the 5–10 years (*p* = 0.005) and >10 years groups of age (*p* < 0.001). Otherwise, the titers of anti-CCP and anti-MCV were similar across all age groups, and there were no statistically significant differences between antibody levels and patients’ ages (Figure 2B).

#### 3.2.3. Association Between Autoantibodies and Disease Duration

To investigate the relationship between autoantibodies and disease duration, RA patients were grouped according to the duration of RA or since disease onset into three periods: <5 years, 5–10 years, and >10 years. At the time of the investigation, the average duration for all patients included in the study was 8.4 ± 6 years. As shown in Table 2, the majority of the patients (42%) had RA duration >10 years, and 35% and 39% had a duration <5 and 5–10 years, respectively. The prevalence and level of each positive antibody (RF, anti-CCP, and anti-MCV) were investigated for each period of duration. The mean disease duration was calculated for the positive-tested antibodies, and the average duration was comparable for all antibodies, approximately 9.3 ± 6.5 years. According to laboratory testing, the positive antibodies were highly present at 5–10 years, followed by >10 years and <5 years in order. However, the existence of each antibody over the periods of 5–10 and >10 years was relatively comparable (Table 5).

For RF and anti-CCP, there were no statistically significant differences (*p* > 0.05) in the prevalences over all periods of RA duration. The frequency of anti-MCV between <5 years and 5–10 years of RA duration showed a statistically significant difference (*p* = 0.004). The presence of positive anti-MCV was ~2-fold higher in patients with a duration of 5–10 years than in those with a duration of less than 5 years (Figure 3A). Overall, the highest incidence of each positive antibody was reported in the 5–10-year RA duration group.

The level of each antibody in each period of RA duration was investigated in the same manner. The average concentration (U/mL) for all antibodies was calculated in each period and compared by one-way ANOVA followed by Tukey’s post hoc test. As shown in Table 5, the highest concentration of each antibody was noticed over the course of 5–10 years. The titers of anti-CCP were similar among the three periods of disease duration (*p* > 0.05). Anti-CCP concentration was relatively reduced for the shortest duration (<5 years) when compared to other durations.

The titers of RF and anti-MCV decreased for RA patients with RA for more than 10 years. The RF titer in this period declined by 1.6-fold as compared to other periods, but overall, there was no statistically significant difference (*p* > 0.05). Regarding anti-MCV, the concentration of anti-MCV antibodies was significantly reduced (*p* = 0.005) in >10-year duration when compared to 5–10 years. The average level of anti-MCV in the period of <10 years was 2.1-fold lesser than in the period 5–10 years of RA duration (Figure 3B).

#### 3.2.4. Effects of Smoking on Autoantibodies

Smoking is considered a risk factor for developing RA and contributes to the production of RA-related antibodies. Of the 116 studied patients, 21 (18%) were smokers (either current or ex-smokers), while 95 (82%) were non-smokers. Although smoking is less common overall in this study, smoking rates among men are increasing far more slowly than those of women. Most of the smoker patients (67%) were over 50 years old, and about 47% had a family history of RA. However, we studied the prevalence and levels of autoantibodies in both smoker and non-smoker patients to evaluate the association between smoking and seropositivity (Table 6). Seropositivity was more frequent in non-smoker patients than smokers, with no significant statistical differences (*p* > 0.05) for RF and anti-CCP antibodies.

As shown in Figure 4A, the existence of RF and anti-MCV was comparable between smokers and non-smokers (*p* > 0.05). For anti-CCP, the positivity rate was more frequent in non-smokers; it was 1.8-fold higher than in smokers, with a significant statistical difference between smokers and non-smokers (*p* = 0.03). Figure 4B shows the average concentration of each antibody in the smokers and non-smokers. Our investigations showed that the titers of each antibody were comparable between smokers and non-smokers. The titers of anti-CCP and anti-MCV were slightly higher in smokers, unlike the RF titer, which was increased in non-smokers. The overall results showed no significant statistical difference (*p* > 0.05) between smokers and non-smokers. According to these findings, smoking does not show a significant effect on the presence and titer of the investigated antibodies in this population.

#### 3.2.5. Family History and Seropositivity

In this study, 25 patients, representing 21.6% of the total population, had a family history of RA. Most of these patients (88%) were above 40 years of age, with an average disease duration of about 9 years. Interestingly, about 40% of RA patients with positive family history in this study were smokers. Table 7 summarizes the prevalence and levels of autoantibodies among RA patients with family history and patients without family history.

The seropositivity was comparable between both groups of patients for anti-CCP and anti-MCV antibodies. The positive anti-CCP and anti-MCV occurred in 52% and 40% of the patients with a positive family history, respectively. However, the incidence of positive RF was higher in RA patients who had no family history (41%) than in those with a family history of RA (24%) but without significant difference (*p* > 0.05). Overall, there was no significant association between seropositivity existence and the presence of family history for RA (Figure 5A). In addition, autoantibody titers were comparable between patients with and without family history, except for RF, which was increased in patients with family history. However, we did not find a significant association (*p* > 0.05) between the concentration of antibodies and the presence of family history for our patients’ groups (Figure 5B).

### 3.3. Exosomal Studies

In the exosome analysis, we classified the RA patients according to their serological findings into seropositive RA (SPRA) and seronegative RA (SNRA). Patients who did not express any positive antibodies were considered SNRA, while those with at least one positive antibody were identified as SPRA. Therefore, the study of exosomes included three groups of samples: SPRA patients, SNRA patients, and HC individuals.

#### 3.3.1. Measurement of Exosome Diameter Dynamic Light Scattering

To confirm that the exosome isolation kit and method used produced a nanoparticle population loaded with exosomes of high purity and typical size, we determined the size and distribution of isolated exosomes using dynamic light scattering (DLS) with a Zetasizer Nano ZS analyzer and assessed their size and morphology using negative stain transmission electron microscopy (TEM). DLS yielded histograms of one and two peaks, with most of the peaks being single. Zetasizer histograms showed how the sizes of exosomes were distributed. Border size distribution was measured and detected in exosomes from the SPRA and HC groups (Figure 6A–C). However, after excluding the large peaks, the mode size of all isolated exosomes was in the expected range of 30–153 nm, and the mean diameter was approximately 76 nm for all sample groups. According to DLS measurements, the exosomes isolated from SPRA and HC samples revealed comparable sizes, with a diameter (mean ± SD) of 96.5 ± 45 nm and 80 ± 45 nm, respectively. In contrast, the exosomes isolated from SNRA patients showed smaller diameters of 52 ± 28 nm. However, the size of SNRA exosomes showed a statistically significant difference compared to SPRA exosomes (*p* < 0.001) and HC exosomes (*p* = 0.004) (Figure 6D).

#### 3.3.2. Characterization of Exosomes by Transmission Electron Microscopy (TEM)

TEM imaging was employed to verify the purity and quality of isolated nanoparticles. In a current study, serum-derived exosomes from patients and HC samples were prepared as detailed in the methodology and subjected to TEM. Likewise, patients’ samples were categorized into SPRA, SNRA, and HC. Using Titan CT TEM, we defined and compared the exosome size and morphological variations between groups. The negative contrast staining revealed very pure exosomes, which were nanosized particles that looked like cups, spheres, and rounds. Our TEM analysis showed that exosomes were more enriched in the SPRA and SNRA groups than in HC samples. Exosomes isolated from SNRA and HC appeared to have a pure mixture and were relatively less homogeneous than SPRA exosomes (Figure 7). TEM images illustrated that all exosome sizes ranged between 30 and 95 nm, and the average size was 47.8 ± 8.5 nm for single exosomes. It is important to note that TEM detects a smaller mean size than DLS, and this appears to be a result of the TEM preparation process, which could cause the exosome to downsize.

#### 3.3.3. Determination of Exosomal Protein Concentration

Protein measurements showed that exosomes isolated from RA patients were more enriched with proteins than the healthy control (HC) individuals. After exosome isolation and lysing, the protein content of each sample was quantified by a Pierce detergent-compatible Bradford assay. The average protein concentrations (mean ± SD, µg/mL) of exosomes derived from SPRA and SNRA samples were comparable (1828 ± 487 µg/mL and 1950 ± 511 µg/mL), respectively. Exosomes isolated from the HC samples had the lowest protein level (1021 ± 375.9 µg/mL). The protein content of HC exosomes was significantly different from SPRA and SNRA exosomes (*p* < 0.001). Figure 8A compares the Bradford assay-determined protein concentrations for RA patient groups and HC.

#### 3.3.4. Identification of Exosomal Proteins by SDS-PAGE

In this study, we ran SDS-PAGE electrophoresis for all exosomes extracted from patients and healthy control samples. The appearance of gel bands showed that there was no significant difference between all exosome preparations. The pattern of Coomassie staining revealed that the protein bands spread almost the same for all exosomes. Several bands between 25 and 250 kDa were identified in the SDS-PAGE analysis using a pre-stained protein marker as a molecular weight standard.

Each exosome lysate exhibited ~14 characteristic bands at various molecular weights. Six bands were major proteins and related to the common exosome protein markers (Figure 8B). Among these obvious proteins, three bands correspond to cytosolic proteins such as Alix (95 kDa), HSP70 (70 kDa), and TGS 101 (46 kDa), while two bands correspond to tetraspanin proteins such as CD9 (23 kDa), CD81 (25 kDa), and CD63 (53 kDa).

#### 3.3.5. Expression of Exosome-Specific Marker: CD9

To confirm the presence of the exosome in our extraction, we investigated the expression of CD9, which is a membranous protein frequently associated with EVs and used as a specific marker for exosomes. CD9 is a member of the tetraspanin family and has a molecular weight of about 23 kDa. The presence of CD9 was evaluated by Western blot in all exosomes derived from RA patients and healthy controls. As described in the methodologies, exosomes were electrophoretically separated and trans-blotted to the PVDF membrane in parallel with a pre-stained protein marker. After staining, bright bands were expressed at the expected size of CD9 (~23 kDa) in all exosomes extracted from RA patients (SPRA and SNRA) and HC samples (Figure 9A). However, exosomes isolated from patients’ groups showed higher intensity than those from healthy control samples.

#### 3.3.6. Detection of Citrullinated Fibrinogen (cFBG) Proteins by WB

ACPAs are specifically and commonly found in the sera of RA patients. cFBG is one of the potential autoantigens for ACPAs in RA patients. To explore the presence of cFBG in serum exosomes, we used monoclonal antibodies specific for cFBG in Western blotting. These antibodies can detect cFBG at an approximate molecular weight of 56 kDa. Therefore, we aimed to investigate the presence of citrullinated protein.

In this study, we performed WB on exosome lysates from seropositive RA patients. In doing so, we excluded the SNRA and HC sample groups and solely carried out immunoblotting on exosomes from the SPRA patient group (Figure 9B). We used monoclonal anticitrullinated fibrinogen antibodies to examine the presence of cFBG in exosomes extracted from SPRA patients (n = 71). Then, the expression of cFBG bands in blotted membranes was correlated to the sample’s seropositivity for anti-CCP. Among the 71 blotted exosomes, 36 samples showed a clear band (cFBG-positive) with a molecular weight of approximately 56 kDa, whereas 35 samples did not (cFBG-negative).

The cFBG-positive (n = 36) exosomes showed a strong correlation with anti-CCP. In this group, all exosomes tested positive for anti-CCP, with 27 patients (75%) being over 50 years old and having RA for more than 5 years. On the other hand, among RA patients with cFBG-negative exosomes (n = 35), only 27 patients (68%) tested positive for anti-CCP, and about 75% were aged above 50 years, while 48% had an RA duration of more than 5 years. However, patients who expressed cFBG on average had 2.6-fold higher levels of anti-CCP than those who did not express cFBG in their exosomes. The average concentration (mean ± SD, IU/mL) of anti-CCP for cFBG-positive exosomes was 727 ± 389 IU/mL, compared to 273 ± 383 IU/mL for cFBG-negative exosomes (Figure 9C).

## 4. Discussion

This study investigated 116 RA patients, with an 8:1 female-to-male ratio. In Saudi Arabia, there are no adequate data available about the exact prevalence of RA in the Saudi population. According to several past studies, the prevalence of the disease among the Saudi population is about 0.1–0.22%, with three to four times higher rates in women than men, which reflects the high risk of the condition worsening [6]. However, recent reports suggest that the incidence of RA is increasing, and the number of RA patients is expected to stand at around 250,000 cases in the near term, i.e., about 1.2% of the Saudi population [7,68]. For all participants, the average duration of RA is 8.4 ± 6 years, while for 66% of the population, the average disease duration is more than 5 years. The mean age for RA patients in this study was 51.3 ± 11.5 years, with 54% of females and 67% of males being above 50 years of age. This is in line with most data, which suggest that RA development occurs in those over the age of 50 [26,69,70].

Our reported female/male ratio is significantly higher than that estimated either globally or within Saudi Arabia, and we thought that was a limitation due to the small number of male patients (12), which represents 10% of the enrolled patients. In Saudi Arabia, many studies on RA patients have been performed in the last decade. A study conducted by Almoallim et al. showed that the female-to-male ratio in a cohort of 433 RA patients was 3:1, with a mean age of 49 ± 11 years [71]. However, women in Latin America reported the highest global prevalence of RA, with a female/male ratio of 6–7:1 in Chile [72], 6:1 in Argentina [73], and 5.2:1 in Colombia [74].

A recent study by Alharbi et al. [75] included 210 patients with a mean age of 46 ± 11 and reported a prevalence of females with a ratio of about 4:1. Interestingly, a study conducted by Aseel et al. demonstrated a female-to-male ratio of 8:1, which included 438 RA patients, which is similar to our findings [76]. However, some population-based studies have reported a higher incidence of RA in females, with a ratio as high as 9:1 [77,78].

Rheumatoid arthritis (RA) is a chronic disease that results from the interaction of multiple genetic, environmental, and lifestyle factors. ACPAs are produced as an autoimmune response for abnormal citrullination reactions. However, ACPAs are specific and predictive biomarkers for RA, and they can be detected in patients’ sera up to 10 years before the clinical RA onset. But, in fact, many RA patients might lack these antibodies. Therefore, RA diagnosis can be classified into seronegative RA (SPRA) and seropositive RA (SNRA) [79].

The association between RA-related autoantibodies and factors such as gender, age, disease duration, and smoking status has been investigated by many previous studies. The influence of age and gender on the development of RA has been described by several global studies [26,80,81]. In the current study, 71 out of 116 patients (61%) had SPRA, and the seropositivity percentages in men were higher than those in women. However, estimations of seropositivity among RA patients have reported inconsistent percentages across a retrospective study [82] and a comparative observational study [83]. Our data showed that about 75% of males had SPRA, while 60% of females had SPRA. Furthermore, the titers for anti-CCP and anti-MCV were higher in males than in females, without significant difference (*p* < 0.05). These findings are consistent with the recent systematic review and meta-analysis study in 2022, which searched databases for eighty-four studies, including 141,381 RA patients, and found that seropositivity is more associated with males than females [22]. However, our results differ from those of a recent cross-sectional study that found females represented 76% of SPRA patients (*p* < 0.001) [84].

Despite the occurrence of ACPA-positive patients in our work increasing gradually with patients’ age, there was no significant association for anti-CCP prevalence with age. For anti-MCV, we found it was significantly frequent in patients above 50 years of age. However, the antibody levels across age groups were almost similar or slightly decreased for both ACPAs (*p* < 0.05). This is supported by a previous study that found a non-significant decrease in anti-CCP levels in older RA patients [85]. The findings of our study showed that SPRA patients are older (52.1 ± 11.2 years) than SNRA patients (50.2 ± 11.9), without significant differences (*p* = 0.39). There are some parallels and some differences between the findings in the literature and ours. Our findings are in line with previous studies that demonstrated that the prevalence of ACPA-positive RA was more frequent at older age, above 50 years [86] and 40 years [85]. Conversely, older age has been found to be strongly associated with SNRA than SPRA (54 ± 11 years vs. 43 ± 14 years; *p* < 0.001) [84]. Furthermore, a multicohort study conducted by Boeters et al. showed that ACPA-negative RA was more associated with older age than was ACPA-positive RA [87].

Our current work also highlights the contribution of RA duration to ACPAs. The 5–10-year duration period reflects the highest incidences and levels for both anti-CCP and anti-MCV. Despite this, we did not find a significant association between positive anti-CCP and RA duration either for prevalence or for antibody levels. A recent observational study showed that the higher incidence of ACPA-positive antibodies was associated with RA duration < 10 years, followed by 5–10 years and <5 years, respectively [88]. A previous cross-sectional study showed that positive anti-CCP was more prevalent in patients with RA duration > 10 years (79%) than in those with duration less than 10 years (62%). This study also revealed that anti-CCP levels were comparable between the two periods of duration [89]. We found that the prevalence of anti-MCV antibodies increased after 5 years of duration, whereas high levels were associated with the 5–10-year duration period. The assessment of anti-MCV by Poulsom and Charles showed that the prevalence of anti-MCV was higher in short-duration RA than in long-duration RA [90].

Smoking is considered a known risk factor for rheumatoid arthritis development [91]. A case–control Myeira study by Abqariyah et al. confirmed the association between smoking and ACPA positivity (64%) in a Malaysian RA population [92]. The current study did not find a relationship between smoking and ACPAs incidence. On the contrary, seropositivity, whether for anti-CCP or anti-MCV, was more frequent in non-smokers than smokers. Our estimations revealed that smoking is responsible for less than 40% of ACPA-positive RA in smokers compared to more than 50% in non-smokers. This is in line with data from the Swedish Epidemiological Investigation of Rheumatoid Arthritis (EIRA) that showed that smoking is responsible for 35% of ACPA-positive RA [93]. In this experiment, smoker RA patients showed a non-significant increase in ACPA titers. However, a previous cross-sectional analysis demonstrated a significant association between smoking and the high titers of ACPA [94].

Our correlation between family history and seropositivity showed no significant association. We found that patients with no family history have a similar tendency to seropositivity as those with positive family history, even for prevalence or concentration. For instance, Diane et al. suggested that the heritability of ACPA-positive and ACPA-negative RA is comparable (65%) [95]. Nevertheless, a heritability of around 50% for ACPA-positive RA and approximately 20% for ACPA-negative RA was shown to be compatible with familial hazards in register-based, nested, case–control research conducted in the Swedish population [96]. The titers of anti-CCP and anti-MCV for RA patients in our work showed no significant differences between those with family history and without. This is not in accordance with the findings of Kim et al., who found that the high levels of anti-CCP and anti-MCV were correlated with the presence of family history [97].

Exosomes are the smallest and most extensively researched EVs that participate in various biological processes and disorders. In the current work, serum-derived exosome preparations of RA patients and HC were validated for morphology and protein content. Negative-staining TEM is the most popular technique for visualizing exosomes [98,99,100]. Using TEM, we verified that all preparations had the typically described morphology of exosomes and were within the expected size range (30–150 nm) [101,102,103]. However, DLS measurements showed that SNRA exosomes were smaller than SPRA exosomes (*p* < 0.001) and HC exosomes (*p* = 0.004). Furthermore, the exosome fraction from SPRA samples showed significant heterogeneity when compared to exosomes from SNRA and HC samples. Exosomes are known to have a very diverse population, and it is challenging to isolate exosomes because of their heterogeneity in size, composition, and function [104,105].

Western blot (WB) analysis is another method of exosome confirmation that looks at preparation purity and the presence of exosome markers. Tetraspanins (CD9, CD63, and CD81) are among the most widely used exosome markers; nevertheless, these proteins have also been found to be expressed positively in other EV types [106]. These proteins have been discovered to be highly enriched in EVs as opposed to originating cells [107]. CD9 is a 25 kDa membranous protein that can be detected by WB. Yue-Ting et al. found that exosomes have an equivalent or even higher level of CD9 than the source cells [108]. In this experiment, all exosome samples derived from patients and controls expressed clear CD9 bands with variable intensity. We found that the intensity of CD9 bands in HC samples is not as strong as in patient samples.

Antibody-based techniques like ELISA, immunohistochemistry, and Western blotting are widely used to detect citrullination with high sensitivity and specificity [109]. But these techniques cannot produce a large-scale analysis and reliable localization of citrullinated sites comparable to that of mass spectrometry (MS) analysis. However, citrullination profiling remains a significant difficulty for mass MS-based approaches despite its increasing effectiveness in many post-translation modification PTM-related investigations [110]. In the current study, we used an anti-CCP commercial kit for the serological investigation of ACPAs, which is an ELISA-based immunoassay. The anti-CCP test was generated by screening peptide libraries containing millions of cyclic peptides to produce highly purified synthetic peptides that serve as antigens [111]. However, some ELISA assays designated for testing anti-CCP do not specify the target antigen. Few studies have used anti-CCP assays that specified the type of citrullinated protein, such as collagen [112], filaggrin [113], and fibrinogen [114].

Fibrinogen and its modified (citrullinated) form have been verified to be the most favored autoantigens for ACPAs in RA patients. Citrullinated fibrinogen (cFBG) epitopes play a role in triggering the autoimmune response in RA patients and contribute to synovitis and bone destruction [65,115]. With regards to EVs, research on exosomes’ role in RA is still in its early stages; a small number of studies have shown that RA patients have aberrant exosome expression [116]. It has been suggested that exosomes may contribute to joint inflammation in RA patients because they can transport autoantigens and mediators between distant cells [117].

Recent proteomic studies were performed on exosomes extracted from the plasma and serum of RA patients; these studies demonstrated distinct protein profiles in the purified exosomes, but no citrullinated proteins were identified [63,118]. Despite the difficulty in identifying citrullinated proteins in exosomes, a previous study compared the proteomic content of exosomes extracted from synovial fluid of RA and osteoarthritis (OA) patients. Skriner et al. were able to detect citrullinated forms of fibrinogen, including fibrin alpha-chain, fibrin beta-chain, fibrinogen beta-chain precursor, and fibrinogen D fragments, in the purified synovial exosomes [67].

In our work, we extracted exosomes from RA patients’ sera, and we confirmed the identity of these microparticles via TEM and DLS. Since cFBG is the best candidate antigen for ACPAs, the goal of our exosome analysis was to explore the presence of cFBG in these microparticles. We used WB to investigate the presence of cFBG in serum-derived exosomes from RA patients, particularly those with positive ACPAs. Several experiments have indicated the contribution of citrullinated proteins in RA pathogenesis. These works investigated cFBG in the serum [119,120,121,122] and synovium of RA patients [66,123,124]. However, to the best of our knowledge, this is the first study exploring the cFBG in exosomes extracted from RA patients’ sera.

So, we investigated whether cFBG is present in serum-derived exosomes from people with RA, especially those who have anti-CCP antibodies that are positive (SPRA). For this purpose, we performed WB using monoclonal anticitrullinated fibrinogen antibodies. Our results showed that 36 of the 71 SPRA patients in this study expressed cFBG with clear bands at a molecular weight of about 56 kDa in their blotted exosomes. The other 35 SPRA patients did not express cFBG.

The immunogenic proteins are not specified in the commercial ACPA ELISA tests, such as anti-CCP. Most CCP tests incorporate multiple reactive citrullinated peptides derived from the most autogenic proteins, such as fibrinogen, vimentin, collagen type II, and α-enolase, which are selected from dedicated synthetic peptide libraries [125,126]. Furthermore, Wegner et al. demonstrated that human fibrinogen and α-enolase are the prominent proteins citrullinated by *P. gingivalis* at the site of periodontitis [127]. As mentioned above, about 50% (36 patients) of SPRA patients in the current study presented cFBG in their exosomes. These findings are similar to those of Xiaoyan et al., who found that the presence of anti-cFBG and the formation of immunological complexes containing citrullinated fibrinogen trigger the complement cascade and contribute to the pathophysiology of RA in one-half of anti-CCP+ RA patients [65]. After correlation with seropositivity, our findings showed that all 36 cFBG-positive exosomes had a positive anti-CCP, while among the 35 cFBG-negative exosomes, 27 patients tested negative for anti-CCP. To explain this, we suggest that these samples may have included citrullinated proteins other than cFBG. Furthermore, we found that the titers of anti-CCP for cFBG-positive exosomes were significantly higher than those for cFBG-negative exosomes. These outcomes give us confidence that autoantibodies against cFBG seem to be the most effective serological criterion for RA diagnosis. Although 75% of patients with cFBG-positive exosomes were above 50 years of age and had RA for more than 5 years, we could not conclude a specific association of cFBG with either patients’ age or disease duration.

The study has its limitations. The number of HC remains small at 35, which can be attributed to our inclusion criteria. We enrolled healthy individuals, and we ensured they were free of family history of immune diseases; in addition, they matched the demographics of the patients. Another limitation is the lack of a genetic background that is strongly associated with autoantibodies (ACPAs). To address this, we investigated the family history of the patients. RA is an inheritable condition, and the familiar risk contributes to 50–60% of RA cases.

## 5. Conclusions

The data presented by this study showed that SPRA is more frequent than SNRA in the Saudi population, with a higher prevalence in men than women. The female-to-male ratio is 8:1, which is higher than that globally reported in Latin America (5–7:1). Further studies are needed to investigate the spike in the number of women affected by the disease. We can conclude that patients’ age and disease duration contribute to the autoantibodies, particularly RF and anti-MCV, whereas smoking and family history had no effects on autoantibodies. Understanding the role of autoantibodies in RA can be improved by conducting in-depth research on autoantibodies in conjunction with other factors, including genetic, cellular, lifestyle factors, and infections related to RA and autoimmune disease, such as periodontitis and *Helicobacter pylori* (*H. pylori*) infection. Our study of serum-derived exosomes revealed that half of the ACPA-positive exosome samples contained cFBG, which greatly contributes to RA progression. Exploring the detailed mechanisms of exosomes in RA pathological changes will help to screen and identify potential therapeutic targets.

## Figures and Tables

**Figure 1 antibodies-14-00010-f001:**
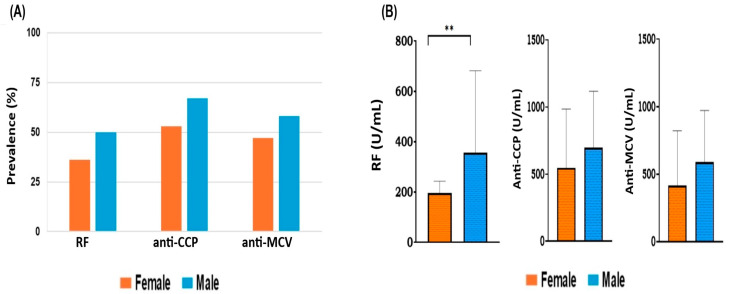
(**A**) Prevalence of positive (RF, anti-CCP, and anti-MCV) in female and male RA patients. The chi-square test was used to compare the frequency of each autoantibody between males and females, and there were no statistically significant differences (*p* > 0.05). (**B**) Relationship between the genders and the level of each antibody. Each bar represents the mean, and error bars correspond to 95% confidence intervals. Using unpaired *t*-test, the concentration of RF showed a statistically significant difference between females and males (** *p* = 0.004). Abbreviation: RF = Rheumatoid factor, anti-CCP = anti-cyclic citrullinated peptide, anti-MCV = anti-mutated citrullinated vimentin.

**Figure 2 antibodies-14-00010-f002:**
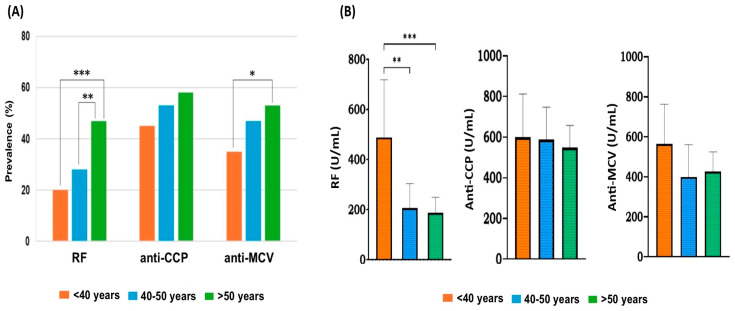
(**A**) Frequency of autoantibodies among age groups of RA patients. Using chi-square test, significant differences are shown for RF incidence between <40 and >50 years groups (***) and between 40–50 and >50 years groups (**). The anti-MCV prevalence showed a statistically significant difference between <40- and >50-year groups (*). (**B**) Titers of antibodies among age groups. The one-way ANOVA followed by Tukey’s post hoc test was used to determine the differences in means. Each bar represents the mean, and error bars correspond to 95% confidence intervals. The statistically significant difference in RF level between age groups is shown, where *** *p* < 0.001; ** *p* = 0.005.

**Figure 3 antibodies-14-00010-f003:**
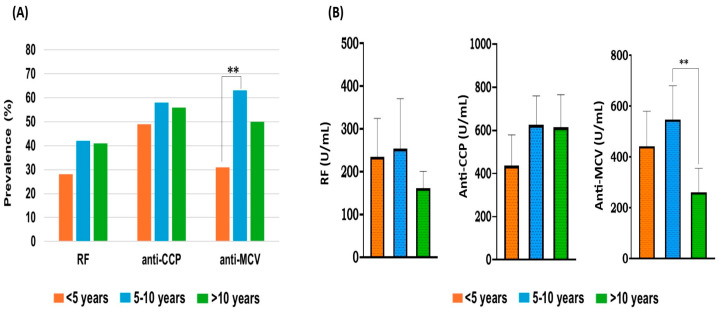
(**A**) Prevalence of autoantibodies in association with the periods of RA duration. No significant difference (*p* > 0.05) for RF and anti-CCP in all periods. The anti-MCV prevalence showed a statistically significant difference (**) between <5- and 5–10-year groups. (**B**) Association between antibody levels and the duration of RA. Each bar represents the mean, and the error bars correspond to 95% confidence intervals. One-way ANOVA followed by Tukey’s post hoc test was used to compare the means across each period. The statistically significant difference in the means of anti-MCV level is displayed (** *p* < 0.01).

**Figure 4 antibodies-14-00010-f004:**
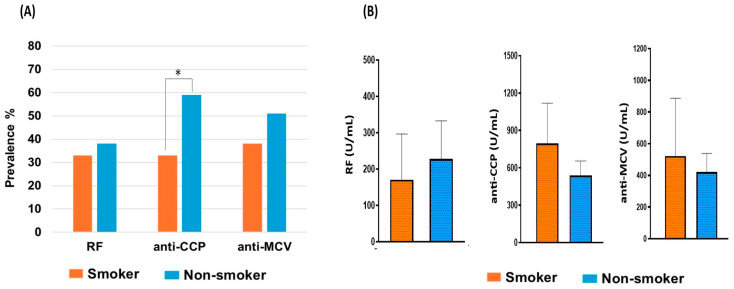
Effect of smoking on autoantibodies. (**A**) The anti-CCP is significantly (*) frequent in non-smokers (*p* = 0.003). No significant difference (*p* > 0.05) for the prevalence of RF and anti-CMCV in both groups. (**B**) No impact of smoking on the titer of positive antibody (*p* > 0.05). The comparison was performed using unpaired *t*-test; each bar represents the mean, and error bars correspond to 95% confidence intervals.

**Figure 5 antibodies-14-00010-f005:**
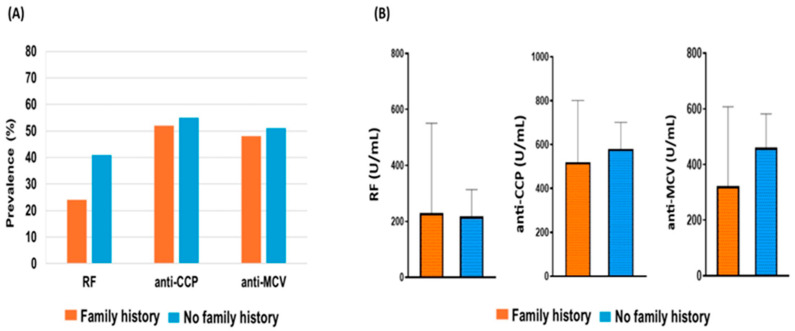
Association between family history and seropositivity. (**A**) Autoantibodies prevalence in RA patients with and without family history showed no significant differences (*p* > 0.05). (**B**) No relationship between family history and the titer of a positive antibody. Each bar represents the mean, and error bars correspond to 95% confidence intervals. An unpaired *t*-test was used to compare the difference in means (*p* > 0.05).

**Figure 6 antibodies-14-00010-f006:**
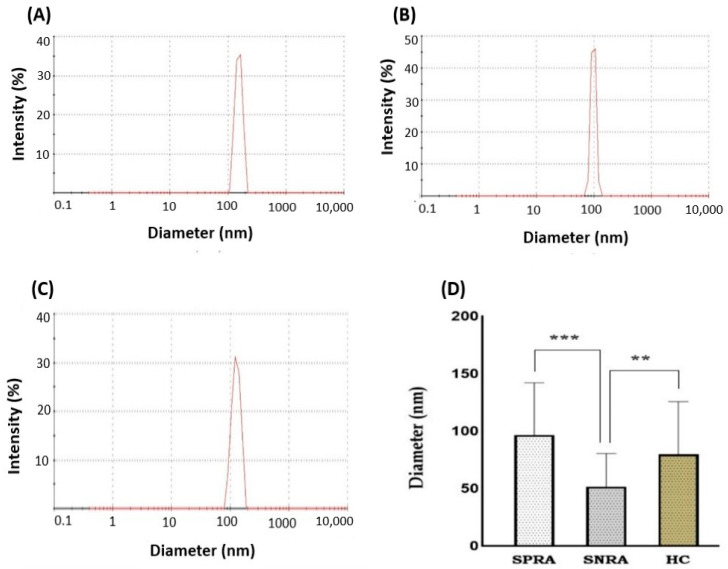
Summary of exosome morphology by DLS. Zetasizer histograms represent the size distribution by intensity. (**A**) SPRA; (**B**) SNRA; (**C**) HC. (**D**) Exosomes isolated from SNRA patients revealed a significantly small diameter when compared to SPRA (***) and HC samples (**) (SPRA: seropositive RA patients; SNRA: seronegative RA patients; HC: healthy control individuals).

**Figure 7 antibodies-14-00010-f007:**
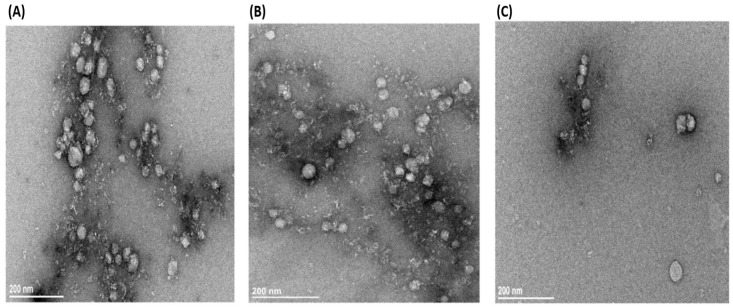
TEM images of exosomes using negative staining. Exosomes showed round-shaped vesicles with diameters less than 100 nm; scale bar = 200 nm. Representative TEM micrograph of exosomes isolated from (**A**) SPRA, (**B**) SNRA, and (**C**) HC (SPRA: seropositive RA patients; SNRA: seronegative RA patients; HC: healthy control individuals).

**Figure 8 antibodies-14-00010-f008:**
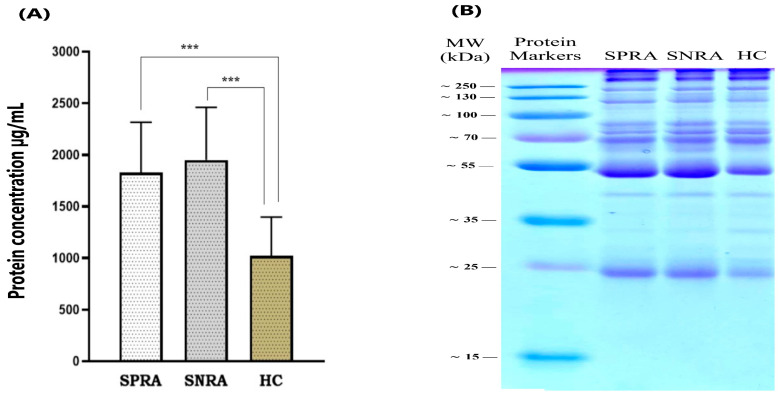
(**A**) Protein content of exosome lysates determined by the Pierce detergent-compatible Bradford assay. Significant differences in exosomes content of protein between patients and HC samples (*** *p* < 0.001). (**B**) Coomassie blue-stained gel showing protein band separation for exosome lysates from patient and healthy control samples (SPRA: seropositive RA patients; SNRA: seronegative RA patients; HC: healthy control individuals).

**Figure 9 antibodies-14-00010-f009:**
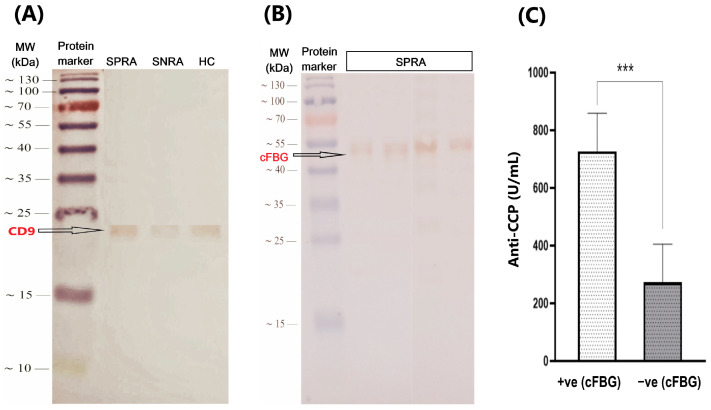
(**A**) Representative Western blotting of CD9 in exosome sample groups: SPRA, ANRA, and HC. (**B**) Western blot analysis for the expression of cFBG in exosome lysates that were extracted from SPRA patients. (**C**) Variation of anti-CCP levels between SPRA patients according to the expression of cFBG in their exosome lysates (*** *p* < 0.001) (SPRA: seropositive RA patients; SNRA: seronegative RA patients; HC: healthy control individuals; cFBG: citrullinated fibrinogen).

**Table 1 antibodies-14-00010-t001:** Demographic characteristics of RA patients and healthy controls.

Characteristic	RA Patients	HC Individuals	*p*-Value
	(N = 116)	(N = 35)	
Gender			
Female, n (%)	104 (89.7%)	28 (80%)	0.13
Male, n (%)	12 (10.3%)	7 (20%)	
Age (mean ± SD, years)	51.3 ± 11.5	47 ± 11.6	0.05
Age range, years	(18–70)	(23–66)	
Age distribution, n (%)			
<40 years	20 (17.2%)	9 (25.7%)	0.26
40–50 years	32 (27.6%)	11 (31.4%)	0.7
>50 years	64 (55.2%)	15 (42.9%)	0.2
Occupation, n (%)			
Worker	26 (22.4%)	11 (31.4%)	0.28
Retired	7 (6%)	2 (5.7%)	0.94
Housewife	81 (69.8%)	20 (57.1%)	0.16
Student	2 (1.7%)	2 (5.7%)	0.2
Smoking, n (%)			
Smoker	21 (18.1)	NA	
Non-smoker	95 (81.9)	NA	
Family history of RA, n (%)			
Yes	25 (21.5)	5 (14.3)	0.35
No	91 (78.5)	30 (85.7)	

Abbreviation: RA = Rheumatoid arthritis. HC = Healthy control individuals.

**Table 2 antibodies-14-00010-t002:** Demographic and laboratory characteristics of RA patients.

Characteristic	All	Female	Male	*p*-Value
	Patients			F/M
Number of patients, [n (%)]	116 (100)	104 (89.7)	12 (10.3)	
Age, mean ± SD (Years)	51.3 ± 11.5	51.3 ± 11.6	51.7 ± 11.3	0.96
Age group, [n (%)]				
<40 years	20 (17.2)	18 (17.3)	2 (16.7)	0.9
40–50 years	32 (27.6)	30 (28.8)	2 (16.7)	0.37
>50 years	64 (55.2)	56 (53.9)	8 (66.6)	0.4
Disease duration, mean ± SD	8.4 ± 6	8.7 ± 6.2	6.1 ± 3.8	0.16
Disease duration, n (%)				
<5 years	39 (33.6)	35 (33.7)	4 (33.3)	0.98
5–10 years	43 (37.1)	36 (34.6)	7 (58.4)	0.11
>10 years	34 (29.3)	33 (31.7)	1 (8.3)	0.09
Smoking, [n (%)]				
Smoker	21 (18.1)	17 (16.3)	4 (33.3)	0.15
Non-smoker	95 (81.9)	87 (83.7)	8 (66.7)	
Family history of RA, n (%)				
Yes	25 (21.6)	23 (22.1)	2 (16.7)	0.66
No	91 (78.4)	81 (77.9)	10 (83.3)	
Positive autoantibody, n (%)				
RF-positive	43 (37.1)	37 (35.6)	6 (50)	0.33
Anti-CCP-positive	63 (54.3)	55 (52.9)	8 (66.7)	0.36
Anti-MCV-positive	56 (48.3)	49 (47.1)	7 (58.3)	0.46

**Table 3 antibodies-14-00010-t003:** The prevalence and levels of positive antibodies among females and males.

Autoantibody	Female	Male	*p*-Value
	N = 104	N = 12	
RF + ve, n (%)	37 (36)	6 (50)	0.33
RF Mean ± SD, (IU/mL)	196.8 ± 236.6	356.4 ± 512.8	0.004 *
Anti-CCP + ve, n (%)	55 (53)	8 (67)	0.36
Anti-CCP (Mean ± SD, (IU/mL))	547.8 ± 437.8	697.1 ± 420.6	0.37
Anti-MCV + ve, n (%)	49 (47)	7 (58)	0.46
Anti-MCV Mean ± SD, (IU/mL)	415.2 ± 406.6	589.2 ± 383.8	0.98

* *p* = 0.004.

**Table 4 antibodies-14-00010-t004:** Prevalence and level of positive antibodies among age groups of RA patients.

Autoantibody	<40 Years	40–50 Years	>50 Years	*p*-Value
	N = 20	N = 32	N = 64	
RF + ve, n (%)	4 (20)	9 (28)	30 (47)	
Mean ± SD, (IU/mL)	488.6 ± 490.9	206.8 ± 271.3	186.8 ± 251.0	<0.001
Anti-CCP + ve, n (%)	9 (45)	17 (53)	37 (58)	
Mean ± SD, (IU/mL)	599.2 ± 454.5	587.7 ± 444.4	549.3 ± 437.9	0.87
Anti-MCV + ve, n (%)	7 (35)	15 (47)	34 (53)	
Mean ± SD, (IU/mL)	564.1 ± 423.9	399.3 ± 445.8	427.4 ± 389.5	0.34

**Table 5 antibodies-14-00010-t005:** Prevalence and level of positive autoantibodies over the period of RA duration.

Autoantibody	<5 Years	5–10 Years	>10 Years
	N = 39	N = 43	N = 34
RF, n (%)	11 (28)	18 (42)	14 (41)
Mean ± SD, (IU/L)	234.3 ± 277.6	254.2 ± 379.7	161.9 ± 114.0
Anti-CCP, n (%)	19 (49)	25 (58)	19 (56)
Mean ± SD, (IU/L)	437.6 ± 438.5	627.0 ± 432.3	616.7± 430.9
Anti-MCV, n (%)	12 (31)	27 (63)	17 (50)
Mean ± SD, (IU/L)	440.9 ± 428.9	546.5 ± 435.3	260.2 ± 273.6

**Table 6 antibodies-14-00010-t006:** Comparison of positive autoantibodies among smokers and non-smoker RA patients.

Autoantibody	Smokers	Non-Smokers	*p*-Value
	N = 21	N = 95	
RF, n (%)	7 (33)	36 (38)	0.7
Mean ± SD, (IU/mL)	170.9 ± 135.7	228.4 ± 308.5	0.63
Anti-CCP, n (%)	7 (33)	56 (59)	0.03 *
Mean ± SD, (IU/mL)	794.8 ± 350.9	538.3 ± 438.9	0.14
Anti-MCV, n (%)	8 (38)	48 (51)	0.3
Mean ± SD, (IU/mL)	522.1 ± 435.7	422.8 ± 402.3	0.52

* *p* < 0.05.

**Table 7 antibodies-14-00010-t007:** Association between seropositivity and family history.

	Positive Family History	Negative Family History	*p*-Value
	N = 25	N = 91	
RF, n (%)	6 (24)	37 (41)	0.13
Mean ± SD, (IU/mL)	229.4 ± 306.1	192.7 ± 292.9	0.58
Anti-CCP, n (%)	13 (52)	50 (55)	0.79
Mean ± SD, (IU/mL)	519.4 ± 467.5	579.1 ± 430.5	0.55
Anti-MCV, n (%)	10 (40)	46 (50)	0.35
Mean ± SD, (IU/mL)	322.5 ± 398.1	372.3 ± 389.7	0.57

## Data Availability

The datasets used and analyzed in this study are available from the corresponding author upon reasonable request.

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
