# Peer review of "Exploring Anticitrullinated Antibodies (ACPAs) and Serum-Derived Exosomes Cargoes"

_2073-4468, 2025, doi:10.3390/antib14010010_

Round 1

Reviewer 1 Report

Comments and Suggestions for Authors

Alghamdi et al studies APCA antibodies in Saudi patients with RA. There is two parts in this study. First relationship of APCA antibodies and clinical parameters. Second, cFBG studies in exosomes. Authors need to do find the relationship between first and second findings.

1. The conclusion of the abstract is not properly explained. It should be concise from the current study only. 

1. The objective of the study is not clear.

2. The Total Exosome Isolation (TEI) kit is mentioned that it is indicated for usage in plasma. However, this study was conducted in serum (line 193). How is it working?

3. Association between disease activity (DAS28) and ACPA well documented in the literature. What about their relationship in this cohort?

4. What is the clinical utility of the detection of cFBG in exosomes? What do authors try to conclude from the study? Why is cFBG studied only in seropositive patients, and what proportions of seronegative patients have cFBG?

5. Is there any correlation between APCA and cFBG levels in these patients?

6. The conclusion is vague and should be clear and precise.

Author Response

General comment: Alghamdi et al studies APCA antibodies in Saudi patients with RA. There is two parts to this study. First relationship of APCA antibodies and clinical parameters. Second, cFBG studies in exosomes. The authors need to find the relationship between the first and second findings.

Response:  Thank you very much for pointing this out. We agree with you that this point needs to be clarified. Therefore, we have described the relation between those two parts in the study's objective. Please refer to the abstract section in the revised manuscript (page 1, lines 25-28)

Comments 1-1: [The conclusion of the abstract is not properly explained. It should be concise from the current study only]

Response 1:  Thank you for your comment. We have revised the conclusion of the abstract to make it clearer in the revised manuscript (page 1, lines 37-42)

Comments 1-2: [The objective of the study is not clear]

Response 2:  Thank you for your helpful comment. We agree with this comment. Therefore, we have added a clear objective of the study in the revised manuscript (Abstract section, page 1, lines 25-28).

Comments 1-3: [The Total Exosome Isolation (TEI) kit is mentioned that it is indicated for usage in plasma. However, this study was conducted in serum (line 193). How is it working?]

Response 3:  Thank you for your comment and question. First, we checked the paragraph for this comment (page 6, line 260) about the kit used for exosome isolation, and it is accurately mentioned that it is used for serum. Second, a Total Exosome Isolation (TEI) kit is used to isolate exosomes from serum by means of precipitation and pelleting. The principle of the TEI kit is based on a precipitation agent known as the hydrophilic polymer polyethylene glycol (PEG).  

Comments 1-4: [Association between disease activity (DAS28) and ACPA well documented in the literature. What about their relationship in this cohort?]

Response 4:  Thank you for your comment. We agree with you that the assessment of DAS28 would be helpful for the study. However, it is important to note that our study focuses on the association between risk factors and autoantibodies in RA patients. DAS28 is commonly used as a tool for disease monitoring, and it serves as a useful quantitative index for the assessment of RA activity upon treatment strategies. DAS28 was not addressed in our study since we couldn’t retrieve the full therapeutic history of the patients. Another concern, it might divert us from the study's main goal.

Comments 1-5: [What is the clinical utility of the detection of cFBG in exosomes? What do authors try to conclude from the study? Why is cFBG studied only in seropositive patients, and what proportions of seronegative patients have cFBG?]

Response 5:  We thank the reviewer for these questions. Firstly, our study proves that serum-derived exosomes carry cFBG, which are known to be autoantigens in RA. Exosomes, through their role in intracellular communication, significantly contribute to the distribution and presentation of citrullinated protein (cFBG) to the effector cells. We suggest that new approaches to RA treatment can be achieved by employing exosomes as potent anti-inflammatory and targeted drug delivery agents. Secondly, the production of ACPA is associated with the presence of citrullinated proteins such as cFBG. According to the lab testing, seronegative patients are not expressing autoantibodies (ACPAs), so it is not expected to have any citrullinated protein in their serum; therefore, we limited the exosomal study to seropositive patients.

Comments 1-6: [Is there any correlation between APCA and cFBG levels in these patients]

Response 6:  Thank you for your valuable comment. Indeed, this correlation is described in the study. For more explanation, we have investigated the levels of ACPAs, particularly the positive anti-CCP. As shown in our results, half of the anti-CCP positive samples have cFBG, while others were negative for cFBG. Accordingly, we compare the levels of anti-CCP between +ve cFBG and -ve cFBG. The results show that the level of ACPAs (anti-CCP) is higher in +ve cFBG than in -ve cFBG. Please refer to the revised manuscript (pages 17-18, lines 651-659).

Comments 1-7: [The conclusion is vague and should be clear and precise]

Response 7:  Thank you very much for your comment. We agree with this comment. Therefore, we have revised the conclusion of the study to make it clearer and updated in the revised manuscript (page 22)

Reviewer 2 Report

Comments and Suggestions for Authors

The manuscript presents valuable insights that merit publication in a journal and will undoubtedly attract the interest of specialists in the field of rheumatoid arthritis studies. I would like to commend the authors for their thorough demographic analysis, which demonstrates a precise and methodical approach to the selection of patient groups.

I propose that the article be contingent upon the following slight revisions:

Lines 156-157: Could the authors please specify the source of the technology used for processing blood serum samples? Is it a standard method or derived from another article? If it is from an external source, kindly indicate the reference.

Lines 772-775: The authors discuss the incidence of RA concerning gender differences in Saudi Arabia. Are similar trends observed in other countries? I recommend including this information in the conclusion, specifying relevant countries to enhance the discussion.

Figure Captions: The current figure captions appear somewhat unclear and lack clarity. I suggest that the figures be produced in higher resolution to improve their quality and readability.

Author Response

General comment 2-8: The manuscript presents valuable insights that merit publication in a journal and will undoubtedly attract the interest of specialists in the field of rheumatoid arthritis studies. I would like to commend the authors for their thorough demographic analysis, which demonstrates a precise and methodical approach to the selection of patient groups.

Response: Thank you for your positive and encouraging feedback.

Comments 2-9: [Lines 156-157:  Could the authors please specify the source of the technology used for processing blood serum samples? Is it a standard method or derived from another article? If it is from an external source, kindly indicate the reference]

Response: Thank you for your comment. All processing technologies used in our work are standard methods, including serum preparation and testing. For example, serum was separated from the blood sample using the centrifugation method. The measurement of rheumatoid factor was achieved through the nephelometry technique, whereas anti-CCP and anti-MCV were quantified by ELISA-based technology.

Comments 2-10: [Lines 772-775:  The authors discuss the incidence of RA concerning gender differences in Saudi Arabia. Are similar trends observed in other countries? I recommend including this information in the conclusion, specifying relevant countries to enhance the discussion]

Response: Thank you for your helpful comment. We agree with this comment. Therefore, we have added the suggested content in the revised manuscript (discussion section, page 18, line 685-687) and (conclusion section, page 22, line 860-862)

Comments 2-11: [Figure Captions:  The current figure captions appear somewhat unclear and lack clarity. I suggest that the figures be produced in higher resolution to improve their quality and readability]

Response: Thank you for your nice reminder. We revised most of the figure captions to make them clearer.

Reviewer 3 Report

Comments and Suggestions for Authors

The present paper claims that autoantibodies such as RF and ACPAs are useful tools in RA. The focus has been on cFBG as a serological marker for diagnosing RA. RA patients' sera were enriched in exosomes found to express citrullinated protein, such as 24 cFBG. Serum autoantibodies in one hundred sixteen Saudi RA patients were evaluated in relation to host-related risk factors. Exosomes were extracted from patients' sera and examined for the presence of cFBG using monoclonal antibodies. The authors reported high female-to-male ratio of 8:1, and the seropositive RA (SPRA) was more frequent among included RA patients. The authors concluded that further studies are needed to confirm the increasing rate of RA among Saudi women affected by the disease. The role of autoantibodies in RA can be improved by conducting in-depth research on ACPA in conjunction with other factors, such as genetic, cellular, and even lifestyle factors. Proteomic study of serum exosomes can be performed and validated in larger populations to develop new diagnostic and prognostic markers for RA.

General aspects:

The present paper is well-written, and the data presented in a relevant way. However, RA has a strong association to many infection-induced diseases, as well as the auto antibodies that are risk markers for RA. These associations are not fully described in the present manuscript, which is a limitation of the manuscript. In addition, is the population of Saudi Arabia representative compared with other populations concerning the prevalence of RA-associated auto antibodies? (this information needs to be added also in the introduction). The genetic factor is also strong in diseases like RA. It is known that the Papillon Lefevre Syndrome has an enhance prevalence in Saudi Arabia. This hereditary disease affects proper immune response, which might interfere with the prevalence of auto antibodies? Infections, as well as immune response play a central role in the formation of exosomes and protein citrullination. Further information in the background is needed in the revised manuscript.

Specific suggestion/comments:

Line 45-47      Add information about the prevalence of RA in Saudi-Arabia in relation to the global prevalence. Is the high prevalence in women also seen in other countries.

Line 64-66      Add chronic infection induced inflammation among the etiologic factors.

Line 80             Is the enhanced prevalence autoantibodies in the ages between 40-60 a global phenomenon? Add information.

Line 111-119  Exosomes are also associated with cancer and infection/inflammation. Could these effects be separated from that of RA risk markers or is it linked to each other?

Line 139-140  The number of HC is low. The prevalence of exosomes and auto antibodies is substantial also in the HC group. This is a limitation of the study.

Line 140          How is the HC population checked? Further information is needed.

Line 185-186  Define the commercial reagent for exosome isolation (company, city, country).

Line 201          Add city and country for Gibco.

Line 207          Add city for Malvin Analytical, UK.

Line 229-239 Add city state and country for all mentioned equipment’s.

Line 241           City and state for Hoefer mini VE.

Line 247          Add city and country for Prism 10 for Windows.

Line 283-600 The result section is impressive, with a large number of different analyzes directed to exosomes and different antibodies. I suggest that the organization of this section could be improved. In addition, my question is if these complex analyzes can be implemented in the prognostic and diagnostic strategy for RA?

Line 284.296  Suggest that the demographics for patients and HC should be included in the material and method section describing study population.

Line 334          Add space after Table 3.

Line 350          Add line in x-axis of figure 1 A. Use the same color for female and male in figure 1 B as used in figure 1 A.

Line 368          Was there any significant differences between age groups in table 4? Add p-values.

Line 385          Add line in x-axis of figure 2 A. Use the same color for the different age groups in figure 2 B as used in figure 2 A.

Line 426          Add line in x-axis of figure 3 A. Use the same color for the different age groups in figure 3 B as used in figure 3 A.

Line 456          Add line in x-axis of figure 4 A. Use the same color for smokers and non-smokers in figure 4 B as used in figure 4 A.

Line 481          Add line in x-axis of figure 5 A. Use the same color for family history in figure 5 B as used in figure 5 A.

Line 595          Figure 5 must be figure 9? Check!

Line 620          Add Alharbi et al. 2023 before reference 56.

Line 646-673 It is not clear in the discussion if the authors believe that autoantibodies is a result of RA inflammation or are risk markers induced by inflammation/infections located on other sites of the body

Line 763-765  It seems to be a strong correlation between the prevalence of anti-CCP and cFBG. What will the analyzes of anti cFGB contribute with in relation to the already established analyzes for anti-CCP? An explanation is needed to be included in the revised manuscript

Line 770          Add limitations of the study. Low number of HC controls and the relatively high prevalence of risk markers in the healthy group. This is a limitation for use in clinical prognostic and diagnostic purposes.

Line 780          Include also chronic infections here, like periodontitis and H. pylori presence in the gastric epithelium,

Author Response

General comment 3-12: The present paper is well-written, and the data is presented in a relevant way. However, RA has a strong association to many infection-induced diseases, as well as the autoantibodies that are risk markers for RA. These associations are not fully described in the present manuscript, which is a limitation of the manuscript. In addition, is the population of Saudi Arabia representative compared with other populations concerning the prevalence of RA-associated auto antibodies? (this information needs to be added also in the introduction). The genetic factor is also strong in diseases like RA. It is known that the Papillon Lefevre Syndrome has an enhanced prevalence in Saudi Arabia. This hereditary disease affects the proper immune response, which might interfere with the prevalence of autoantibodies. Infections, as well as immune response, play a central role in the formation of exosomes and protein citrullination. Further information in the background is needed in the revised manuscript.

Response: Thank you very much for your valuable comments. We have gone through your comments carefully and tried our best to address them one by one. We hope the manuscript has been improved accordingly.

Comments 3-13: [Line 45-47 Add information about the prevalence of RA in Saudi-Arabia in relation to the global prevalence. Is the high prevalence in women also seen in other countries]

Response: Thank you for pointing this out. We agree with this comment. Therefore, we have added the suggested content in the revised manuscript (page 2 , line 50-54).

Comments 3-14: [Line 64-66 Add chronic infection-induced inflammation among the etiologic factors]

Response 2: Thank you for pointing this out. We agree with this comment. Therefore, we have added the suggested content in the revised manuscript (page 2 , lines 74-87).

Comments 3-15: [Line 80 Is the enhanced prevalence of autoantibodies in the ages between 40-60 a global phenomenon? Add information]

Response: Thank you for pointing this out. We agree with this comment. To clarify, the prevalence of autoantibodies in this age group (40-60) is not a global phenomenon, but we did highlight it based on recent studies that investigated the association between RA and other factors, including the age of patients. Therefore, we have provided the following citation in the revised manuscript (page 2, lines 100-103) as support:

-          Takanashi, S., T. Takeuchi, and Y. Kaneko, Effects of Aging on Rheumatoid Factor and Anticyclic Citrullinated Peptide Antibody Positivity in Patients with Rheumatoid Arthritis. J Rheumatol, 2023. 50(3): p. 330-334.

-          Nilsson, J., et al., Influence of Age and Sex on Disease Course and Treatment in Rheumatoid Arthritis. Open Access Rheumatol, 2021. 13: p. 123-138.

Comments 3-16: [Line 111-119 Exosomes are also associated with cancer and infection/inflammation. Could these effects be separated from that of RA risk markers or is it linked to each other?]

Response: Thank you for pointing this out. It is a good comment. Therefore, we have added more information about the role of exosomes in linking RA to malignancy and infection. Please refer to the revised manuscript (page 3, lines 144-157)

Comments 3-17: [Line 139-140 The number of HC is low. The prevalence of exosomes and autoantibodies is substantial also in the HC group. This is a limitation of the study]

Response: Thank you for pointing this out. We agree with this comment. Selecting HC for this study was not an easy task. We select the individuals who are not suffering from any significant illness relevant to our study and meet the defined inclusion criteria, which include factors such as age, sex, and occupation, and we make sure there is no family history of the disease in addition to not smoking.

HC samples were subjected to autoantibody testing and exosomal study in the same way as patients' samples. The size, morphology, and protein content of HC exosomes are already described in the result section (Figures 6, 7, 8). The level of exosomes in RA and other autoimmune diseases is known to be significantly higher than in HC and it is already mentioned in the revised manuscript (Page 3, lines 125-126) although we did not address it in the study. HC samples were tested for autoantibodies, and all were negative, which is one of the criteria for choosing healthy individuals. Therefore, we have included HC in the contents of methods section in the revised manuscript (page 5, lines 233, and page 6, line 242).

Comments 3-19: [Line 140 How is the HC population checked? Further information is needed]

Response: Thank you for pointing this out. We agree with this comment. Therefore, we have added more information about this comment in the revised manuscript (page 4, lines 189-195).

Comments 3-20: [Line 185-186 Define the commercial reagent for exosome isolation (company, city, country)]

Response: Thank you for pointing this out. We agree with this comment. Therefore, we have added the suggested content in the revised manuscript (page 6, line 251-252).

Comments 3-21: [Line 201 Add city and country for Gibco]

Response: Thank you for pointing this out. We agree with this comment. Therefore, we have added the suggested content in the revised manuscript (page 6, line 268).

Comments 3-22: [Line 207 Add city for Malvin Analytical, UK]

Response: Thank you for pointing this out. We agree with this comment. Therefore, we have added the suggested content in the revised manuscript (page 6, line 274).

Comments 3-23: [Line 229-239 Add city-state and country for all mentioned equipment]

Response: Thank you for pointing this out. We agree with this comment. Therefore, we have added all suggested content in the revised manuscript (page 7, lines 298-309).

Comments 3-24: [Line 241 City and state for Hoefer mini VE]

Response: Thank you for pointing this out. We agree with this comment. Therefore, we have added the suggested content in the revised manuscript (page 7, line 311-312).

Comments 3-25: [Line 247 Add city and country for Prism 10 for Windows]

Response: Thank you for pointing this out. We believe you are referring to line 272, not line 247 in the original manuscript. We agree with this comment. Therefore, we have added the suggested content in the revised manuscript (page 8, line 343).

Comments 3-26: [Line 283-600 The result section is impressive, with a large number of different analyzes directed to exosomes and different antibodies. I suggest that the organization of this section could be improved. In addition, my question is if these complex analyses can be implemented in the prognostic and diagnostic strategy for RA?]

Response: We appreciate the encouraging and helpful comments. First, regarding your suggestion to improve the result section, we tried our best to reorganize this section. Our decision to divide the results into two sections—one for the correlation between autoantibodies and risk factors and the other for the exosome study—was driven by the order of the experimental procedures. We found that this is more effective in making the study's findings easy to comprehend.

Secondly, our study focuses on studying the relation between autoantibodies and variable factors and exploring the presence of citrullinated protein in exosomes. However, we agree that these analyzes can be implemented in the prognostic and diagnostic strategy for RA. However, anti-CCP and anti-MCV antibodies represent superior markers for the diagnosis and prognosis of RA. Exosome can also be used as a diagnostic marker for RA, and it is more effective in therapeutic strategies for RA [1].

Comments 3-27: [Line 284.296 Suggest that the demographics for patients and HC should be included in the material and method section describing study population]

Response: Thank you for your suggestion. We agree with this comment. Therefore, we have moved this content and table 1 from the results section to the material and method section in the revised manuscript (Pages 4-5, lines 203-225)

Comments 3-28: [Line 334 Add space after Table 3]

Response: Thank you very much for your nice reminder. We agree with this comment. Therefore, we have added space after table 3 in the revised manuscript (page 9, line 396).

Comments 3-29: [Add line in x-axis of figure 1 A. Use the same color for female and male in figure 1 B as used in figure 1 A]

Response: Thank you for pointing this out. We agree with this comment. Therefore, we have modified figure 1 accordingly in the revised manuscript (page 10, line 412).

Comments 3-30: [Was there any significant differences between age groups in table 4? Add p-values]

Response: Thank you for your reminder. Therefore, we have added the p-values in table 4 in the revised manuscript (page 10-11).

Comments 3-31: [Line 385 Add line in x-axis of figure 2 A. Use the same color for the different age groups in figure 2 B as used in figure 2 A]

Response: Thank you for pointing this out. We agree with this comment. Therefore, we have modified figure 2 accordingly in the revised manuscript (page 11, line 447).

Comments 3-32: [Line 426 Add line in x-axis of figure 3 A. Use the same color for the different age groups in figure 3 B as used in figure 3 A]

Response: Thank you for pointing this out. We agree with this comment. Therefore, we have modified figure 3 accordingly in the revised manuscript (page 12, line 488).

Comments 3-33: [Add line in x-axis of figure 4 A. Use the same color for smokers and non-smokers in figure 4 B as used in figure 4 A]

Response: Thank you for pointing this out. We agree with this comment. Therefore, we have modified figure 4 accordingly in the revised manuscript (page 13, line 520).

Note: During our revision for this section, we found entry mistakes in table 6 for the number of smoker and non-smoker patients. We corrected it and highlighted it in the revised manuscript (table 6, page 13). These were just typing errors that didn’t affect the measurements and findings for this section because all calculations and statistics for this topic were done on the accurate values. Also in the same context, we have modified the related sentences (updated in the revised manuscript (page 13, line 498-499).

Comments 3-34: [Add line in x-axis of figure 5 A. Use the same color for family history in figure 5 B as used in figure 5 A]

Response: Thank you for pointing this out. We agree with this comment. Therefore, we have modified figure 5 accordingly in the revised manuscript (page 14, line 545).

Comments 3-35: [Line 595 Figure 5 must be figure 9? Check!]

Response: Thank you very much for your nice reminder. We have corrected the figure number to figure 9 in the revised manuscript (page 18, line 661).

Comments 3-36: [Line 620 Add Alharbi et al. 2023 before reference 56.!]

Response: Thank you for your nice reminder. We have added the suggested content in the revised manuscript (page 18, line 688).

Comments 3-37: [Line 646-673 It is not clear in the discussion if the authors believe that autoantibodies is a result of RA inflammation or are risk markers induced by inflammation/infections located on other sites of the body!]

Response: Thank you for pointing this out. We agree with this comment. Autoantibodies or ACPAs act as effector molecules in RA pathogenesis. The production of ACPAs is induced by the presence of citrullinated proteins that might be produced in different sites of the body as a result of conditions such as oral infection and gut microbiota dysbiosis. Besides their role in RA inflammation, ACPAs have prognostic values in the identification of “at risk” individuals.

Comments 3-38: [Line 763-765 It seems to be a strong correlation between the prevalence of anti-CCP and cFBG. What will the analyzes of anti cFGB contribute with in relation to the already established analyzes for anti-CCP? An explanation is needed to be included in the revised manuscript

Response: Thank you very much for pointing this out. We agree with your comment.  Our focus on cFBG needs clarification. Therefore, we have included further details regarding this aspect in the revised manuscript (page 21, line 832-851)

Comments 3-39: [Line 770 Add limitations of the study. Low number of HC controls and the relatively high prevalence of risk markers in the healthy group. This is a limitation for use in clinical prognostic and diagnostic purposes]

Response: Thank you for your comment. The reviewer is correct. Therefore, we have added the limitations of the study in the revised manuscript (page 22, line 852-858).

Comments 3-40: [Line 780 Include also chronic infections here, like periodontitis and H. pylori presence in the gastric epithelium]

Response: Thank you for pointing this out. We agree with this comment. Therefore, we have included the suggested comment in the revised manuscript (page 21, line 836-837).

References:

  1. Zhang, S., et al., The impact of exosomes derived from distinct sources on rheumatoid arthritis. Front Immunol, 2023. 14: p. 1240747.

Round 2

Reviewer 1 Report

Comments and Suggestions for Authors

Authors statisfactorly addressed all comments. 

Reviewer 3 Report

Comments and Suggestions for Authors

The revised manuscript is substantial improwed and I hav no further comments.